



# A socio-hydrologic framework for understanding conflict and cooperation in transboundary rivers

Yongping Wei[1,#], Jing Wei[2,#], Gen Li[3,#], Shuanglei Wu[1], David Yu[4], Fuqiang Tian[2,*], Murugesu Sivapalan[5]

1 School of Earth and Environmental Sciences, the University of Queensland, Brisbane, 4072, Australia.

2 Department of Hydraulic Engineering, Tsinghua University, Beijing, 100084, China.

3 School of Ecological and Environmental Sciences, East China Normal University, Shanghai, 200241, China.

4 Lyles School of Civil Engineering, Purdue University, West Lafayette, IN, 47907-2051, USA.

5 Department of Geography and Geographic Information Science and Department of Civil and Environmental Engineering, University of Illinois at Urbana-Champaign, Urbana, IL, 61801, USA.

# These authors contributed equally.

* *Correspondence to*: Fuqiang Tian (tianfq@tsinghua.edu.cn)

**Abstract.** Increasing hydrologic variability, accelerating population growth, and resurgence of water resources development projects have all indicated increasing tensions among the riparian countries of transboundary rivers. This article aims to review the existing knowledge on conflict and cooperation in transboundary rivers from a multidisciplinary perspective and propose a socio-hydrological framework that integrates the slow and less visible societal processes with existing hydrological-economic models, revealing the hidden feedbacks between changes in societal processes and hydrological changes. This framework contributes to understanding the mechanism that drives conflict and cooperation in transboundary river management.

## 1 Introduction

There are 310 rivers around the world that cross the boundaries of two or more countries. When reaping the benefits of the river is perceived as a zero-sum game (Baranyai, 2020), riparian countries in the upstream and downstream often experience more tensions than cooperation (Dinar, 2004). Divergent interests that drive such dynamics cover water quantity, water quality, hydropower infrastructure development, flood management, navigation, economic development, environmental issues, climate change consequences, among many others (Milman & Gerlak, 2020; Nordås & Gleditsch, 2007; Rai et al., 2017; Munia et al., 2016). Increasing hydrologic variability under climate change, accelerating population growth and urbanization, and the resurgence of water resources development projects further exacerbate the tensions among the riparian countries of transboundary rivers (De Stefano et al., 2017). Thus, understanding the mechanisms that drive conflict and cooperation is critically important for addressing this globally increasing issue.

Understanding what explains conflict and cooperation that arise in transboundary rivers is by no means a simple challenge. It is no wonder, therefore, that various disciplines have examined what can contribute to conflict and cooperation in transboundary rivers, and in doing so, covered a wide range of factors (Zeitoun et al. 2013; Petersen-Perlman et al., 2017; Fischhendler., 2008; Ho, 2017). Factors have been investigated from a hydrological perspective such as spatial location





(Schmid, 2008), water availability (e.g., Toset et al., 2000; Furlong et al., 2006; Gleditsch et al., 2006), infrastructure development (De Stefano et al., 2017), external water dependency (e.g. Milman & Ray, 2011), climate change (Gleditsch, 2012), and presence of negative transboundary impacts or interlinkages between water and other issues (Schmeier 2014); from

an economic perspective such as commercial trade (Espey and Towfique, 2004; Tir and Ackerman, 2009; Dinar et al., 2015), and economic development level (Priscoli and Wolf 2009); from a cultural perspective such as saliency of the river (Hensel et al., 2008), peacefulness of riparian relationships (Brochmann and Gleditsch, 2012), identity or national values (Allouche 2005), perceived exposure to unilateral overexploitation of the resource (Elhance 1999), and professionals communities (Kibaroglu, 2008); and from a political perspective such as level of democracy (Brochmann and Hensel, 2009), existence of transboundary

treaties (Brochmann, 2012; Wolf et al., 2003a; Tir and Stinnett, 2012; Dinar et al., 2015), relative power of riparian states (Mirumachi and Allan 2007, Zeitoun et al. 2013), behaviour of the regional hegemon (Zeitoun and Warner 2006), domestic political rivalry, political leadership (Dinar 2009, Subramanian et al. 2014), and institutional resilience (De Stefano et al. 2012). While the breadth of factors implies the importance of using a multidisciplinary approach, hydrology, by analysing the possibilities of where, how and when water can be harnessed and utilised, is commonly recognised as the core discipline for

understanding conflict and cooperation in transboundary rivers. However, there is no systemic understanding yet of how these disciplines, in particular the social sciences, have been integrated with hydrology, which compromises theoretical development and practical management in transboundary rivers.

Transboundary rivers are complex adaptive systems of hydrological, environmental, economic, cultural, and political interdependencies among the riparian countries (Newig & Rose, 2020). Conflict and cooperation are, therefore, emergent

phenomena of the system arising from such interactions. Socio-hydrology, by observing and explaining unintended consequences, as emergent phenomena in the co-evolution of coupled human and water systems at basin/regional scale, provides an interdisciplinary systems framework to understand conflict and cooperation in the complex transboundary river system (Sivapalan et al., 2012; Di Baldassarre et al., 2019; Yu et al., 2020). Socio-hydrology is meta-theoretical and thus should be compatible with theories and models used in different constituent disciplines. It requires a conceptual framework

that acts as a 'middle ground' between the meta-level concepts and specific theories and models driven by a particular discipline or context. This framework would provide a general set of variables and relationships between them from which analysts can choose a subset and further specify according to a specific problem and system being investigated (e.g., Ostrom 2009). In this way, the framework would provide a common language to compare different cases and phenomena and a common database. However, such a middle-ground framework in socio-hydrology for studying conflict and cooperation in

transboundary river systems has been largely absent.

Contributing to the filling of this research gap is the objective of this study. This will be done through three steps. First, a review of the existing literature on conflict and cooperation in transboundary rivers from multiple disciplines in social sciences and their integration with hydrology will be provided. This is a precursor to the next step: proposition of a conceptual socio-hydrology framework that will provide a general set of variables and relationships to consider for studying conflict and



cooperation in transboundary river systems. In the final step, the proposed framework is applied to three cases of transboundary rivers to illustrate its potential utility: the Columbia River, the Mekong River, and the Nile River.

## 2 Literature review on conflict and cooperation in transboundary rivers

### 2.1 Understandings from empirical/assessment studies

There are very rich empirical studies on conflict and cooperation in transboundary river management in the form of collated

datasets. Several global databases on conflict and cooperation have been developed. The International Water Event Database (IWED) (Wolf et al., 2003) documents global water events on conflicts and cooperation during 1948–2008. The Transboundary Fresh Water Dispute Database (TFDD) is a database specifically for global and regional assessment on water conflict and resolution processes (Munia et al., 2016). The Water-Related Intrastate Conflict and Cooperation (WARICC) dataset focuses on events of national water dispute among 35 countries in the Mediterranean, the Middle East, and the Sahel from 1997 to

2009 (Bernauer et al., 2012). Various sets of indictors have also been developed to evaluate the level of cooperation and conflict from different perspectives. The Pacific Institute categorizes water conflict events based on the purpose of water control, where water is considered as a "military tool" or a "political tool" (Pacific Institute, 2009). The Water Cooperation Quotient identifies formal agreements, river basin commissions, ministerial meetings, technical projects, joint monitoring of water flows, floods, dams and reservoirs, high political commitment, integration into economic cooperation, and actual

functioning as ten key aspects that facilitate collaborations between two or more countries (Baranyai, 2020; StrategicForesightGroup, 2015). Zeitoun and Mirumachi have developed quantifiable, two-dimensional matrices (Zeitoun & Mirumachi, 2008), then extend them as the Transboundary Water Interaction NexuS (TWINS) that focuses on comparisons of conflict and collaboration among different countries and how they evolve in time (Mirumachi & Allan, 2007). Wolf et al. developed a 15-point "Basins at Risk (BAR) scales" (Wolf et al., 2003) to classify and measure the extent of water conflict

and cooperation. The Integrated Basin at Risk (iBAR) coding system further include inequalities and injustices into consideration (Watson, 2015). Conca (2006) proposed the core normative elements for assessing transboundary governance: the equitable use principle, no-harm principle, sovereign equality and territorial integrity, information exchange, consultation with other riparian states, prior notification, environmental protection, and peaceful resolution of disputes.

These databases provide a global picture of conflict and cooperation events in transboundary rivers from different temporal

and spatial scales and the assessment studies define and measure conflict and cooperation events with various sets of indicators. These studies provide rich descriptions on the phenomena of conflict and cooperation, however, due to no link with process-





based, cause-effect analysis, they have limited ability to reveal the adaptive evolution and predict future trend on conflict and cooperation between riparian countries.

## 2.2 Understandings from multiple disciplines

Hydrological studies have made major contribution to the understanding of conflict and cooperation in transboundary rivers. They include site-specific and topic-specific studies on the impacts of spatial location, water availability, external water dependency, climate change, and infrastructure development in transboundary rivers (De Stefano et al., 2017; Furlong et al., 2006; Nordås & Gleditsch, 2007). Hydrological models have been developed to assess the biophysical consequences of conflict (unilateral action without agreement among riparian countries) and the biophysical possibility of cooperation by simulating

the impact of upstream alternations of water quantity, flow duration, water quality and river morphology on the ecosystems, agriculture, fisheries, energy production and navigation in downstream countries. By analysing the possibilities of where, how and when water can be harnessed and utilised, hydrological understanding forms the biophysical basis for transboundary river management (Newig & Rose, 2020).

Neoclassical economics has dominated the simulation and explanation of human cooperation behaviour. It explains

cooperation in riparian countries from a purely economic perspective, focusing on the tangible outcomes received by these countries, assuming them as rational actors with perfect information about all potential choices and their consequences (Schill et al., 2019). Hydrological models have been integrated with neoclassical economic models to simulate cooperation in transboundary rivers by optimizing the incremental economic benefits under a set of specific societal constraints. Thus, the influences from the social dimension are only considered as residuals from explanations of rational economic behaviour. These

models have been criticized for being overly simplistic, and unable to capture the diversity of human behaviour (Schlüter et al., 2017), and thus fail to reflect the reality of conflict and cooperation in transboundary rivers (Wei et al., 2021).

Behavioural economics emerges through relaxing the unbounded rationality of actors in the neoclassical economic models (Conlisk, 1996). It argues that decision-makers' preferences are not only deeply influenced by their living environments, social norms, traditional cultures, but also by the inability of policymakers to consistently compare outcomes and their 'mental inertia'

over a long timeframe (Schill et al., 2019). In transboundary rivers, whether people choose to cooperate or not relies on one country's expectations on absolute economic benefits, their benefits in previous periods as a reference level, relative to gains compared to other countries, and intangible benefits such as ecological, social, political, or diplomatic benefits. Hydrological models have been integrated with game theory, agent-based modelling, and system dynamic models to simulate conflict and cooperation in transboundary rivers (Yu et al., 2019; Khan et al., 2017; Ding et al., 2016; Sehlke and Jacobson, 2005). Among

the obvious weaknesses in these models, the one highlighted here is that there are constant difficulties in defining, isolating or





accounting for every set of influential factors, potentially minimising the social dimensions on cooperative behaviours by means of anonymous subjects (Futehally, 2014; Ribes-Iñesta et al., 2006).

Institutional economics is another branch of economics, which focuses on the understanding of inter-organizational cooperation by assessing economic performance under different institutional contexts (Schmid, 2008). In transboundary rivers, institutional economics often collaborates with law to examine treaties and agreements to provide confidence and compliance for negotiation and to reduce transaction costs of cooperation (Rees, 2010; Boin and Lodge, 2016; Saleth and Dinar 2004). Some studies argued that institutional incapacity lies at the root of most water conflicts, where rapid biophysical (e.g. unilateral development projects, unanticipated droughts or floods) and socio-economic changes (e.g. population growth, technological development) have outpaced the institutional capacity to absorb these changes (Wolf et al., 2003). Yet, these studies lack support from process-based hydrological understanding. Hydrological models often simulate institutional conditions and transaction costs of cooperation using game theory, which suffer from the same shortcomings as mentioned in behavioural economic studies.

Cognitive psychology and cultural sociology provide a rich understanding of cooperative behaviours from the perspective of social comparison, self-reflection, and mental model of the future (Schlüter et al., 2017). Social psychologists recognise that people are fundamentally different in respect of their social values and personality traits. These values and traits are primary drivers of cooperative motives and choice behaviour, which can have a mixed influence on cooperation in the situation of social dilemma (Bogaert et al., 2012; Hoff & Stiglitz, 2016). Two opposing social value orientations are typically recognized: a pro-self and a pro-social orientation. Pro-socials believe that it is efficient and fair to cooperate, whereas pro-selfs cooperate because they believe that they will be worse off when they do not (Bogaert et al., 2008). Schwartz (1992) and Howat (2019) identify 10 basic values of social motivation including openness to change, conservation, self-transcendence, self-enhancement, conformity, and others, and depict their relationships to each other. The implication of these theories is that to encourage cooperative behaviours may require different approaches. The commonly adopted method for value measure in these disciplines is to observe the actual choice behaviour in a typical small-scale and short-term experiment. A shortcoming of this method is that it is impossible to separate an individual's natural inclination from the situational determinants that also impact behaviour and existing evidence shows that such measure is not unambiguous. Most studies on conflict and cooperation in transboundary rivers from these disciplines are conceptual such as the prominence of water, identity or national values, and perceived exposure to resource overexploitation (Baranyai, 2019; Brochmann & Gleditsch, 2012; Elhance, 1999) and they have not been integrated into hydrological models.

In part due to the salience of equity, sovereignty, diplomacy and national security in transboundary river management, scholars in political science and international relations have made important contributions in understanding cooperative behaviours in transboundary rivers (e.g., Giordano & Wolf, 2003; Munia et al., 2016). Politics is the study of power (Lasswell, 2003). Hydro-politics is characterized by hegemonic configurations in the form of geographical locations, wherein the most powerful riparian





countries have an advantage over their weaker neighbours on water allocation in transboundary rivers and enforce a cooperative agreement (Mirumachi & Allan, 2007; Zeitoun et al., 2011). Another research field is hydro-diplomacy (water diplomacy),

which refers to an approach that seeks to establish or improve cooperation and stability over water use (Milman & Gerlak, 2020). Cooperation is considered as a two-way interaction between domestic politics and international politics, bounded with concerns of sovereignty around core values (the importance of water in national security) and cultural constructions that date back generations (e.g. the religious dimensions of water) (Warner, 2016). Schwartz et al. (2014) and Howat (2019) used eight political values to understand intergroup conflict: equality, civil liberty, self-reliance, free enterprise, military strength, blind

patriotism, law and order, and traditional morality. Both fields argue that transboundary river management is all about "a political process subject to the whims of power" (Zeitoun & Mirumachi, 2008), leaving little room for economic cooperation. It is fully agreed in both fields that hydrological knowledge (hydrology) is the basis, however hydrological models have not been integrated with political or diplomatic understandings.

The physical interdependence and connectivity of transboundary rivers give rise to regional studies that focus on management

integration. Establishing linkages is considered as the basis for benefit-sharing and crucial for cooperation in transboundary river management (Warner, 2016). These studies hold that issue trading and package deals enable greater economic efficiency and the widest possible range of potential benefits covering economic, social, environmental and political gains that cooperation could bring (Wolf, 2010). The rationality underlying this is that conflict and cooperation cannot be understood by neglecting the broader social and political contexts of transboundary riparian countries (Kibaroglu, 2019). While these linkage

concepts can broaden the solutions of cooperation in transboundaryrivers, no empirical study with hydrology models has demonstrated its practicability and it is also recognized that linkage with broader unbounded contexts could make conflict and cooperation even more complex and difficult to manage.

We sum up current understandings on conflict and cooperation in transboundary rivers as follows: hydrological models have been developed to simulate the biophysical consequences of conflict and the biophysical conditions of cooperation in

transboundary rivers. They have been well integrated with neoclassical economic models to simulate the feasibility of cooperation. Recent developments on hydrology by integrating game theory, agent-based models and system dynamics into hydrological models intend to overcome the over-simplicity in neoclassical economic models by capturing the diversity of human behaviours, however, these models are criticized on their weak and inexplicit representation of social dimensions. Psychology and sociology provide rich descriptions on social motives (values) as primary drivers of cooperative behaviours

but encounter difficulties in integrating with hydrological models as most of their findings are qualitative in nature. Similar lack of integration is also found in hydro-politics and hydro-diplomacy, which emphasize the power differences between riparian countries. More importantly, conflict and cooperation have never been understood as an emergent outcome from the





co-evolved processes in a complex and dynamic system (except Lu et al., 2021 on this issue). Therefore, current understandings have limited analytical capacity to reveal the mechanism that drives conflict and cooperation.

## 3 A social-hydrological framework for understanding cooperation and conflict in transboundary rivers

### 3.1 The framework

We develop a meta-theoretical framework to address the knowledges gaps in understanding cooperation and conflict in transboundary rivers identified in the section above. This framework should act as a 'middle ground' between the meta-level concepts and theories from related disciplines as introduced above and specific models driven by a particular context/a specific problem. The 'middle ground' is expressed as a general set of variables and relationships from which analysts can choose a subset and further specify according to a specific problem being investigated. We recognize that transboundary rivers are complex adaptive systems of hydrological, environmental, economic, cultural, and political interdependencies among the riparian countries, thus develop this framework by following the system theory in particular the complex system theory. The development of this framework are also built on the recent advances on understanding the coupled human-environment relationships from social-ecological system (Folke et al, 2005), the Coupled Human and Nature Systems (CHANS) (Liu et al, 2007) and the general social-hydrological framework (Elshafei et al, 2014). Figure 1 presents our proposed social-hydrological framework for identifying the cause and effect of conflict and cooperation in transboundary rivers to improve cooperative management, in which cooperation is considered as a continuum of two opposing ends from extreme conflict to full cooperation. This framework extends the existing understanding of cooperation from integrated neoclassical economics and hydrology (as the fast processes indicated in upper part of Figure 1) to inclusion of the willingness to cooperate, a hidden variable representing the societal processes (as the slow processes in lower part of Figure 1).





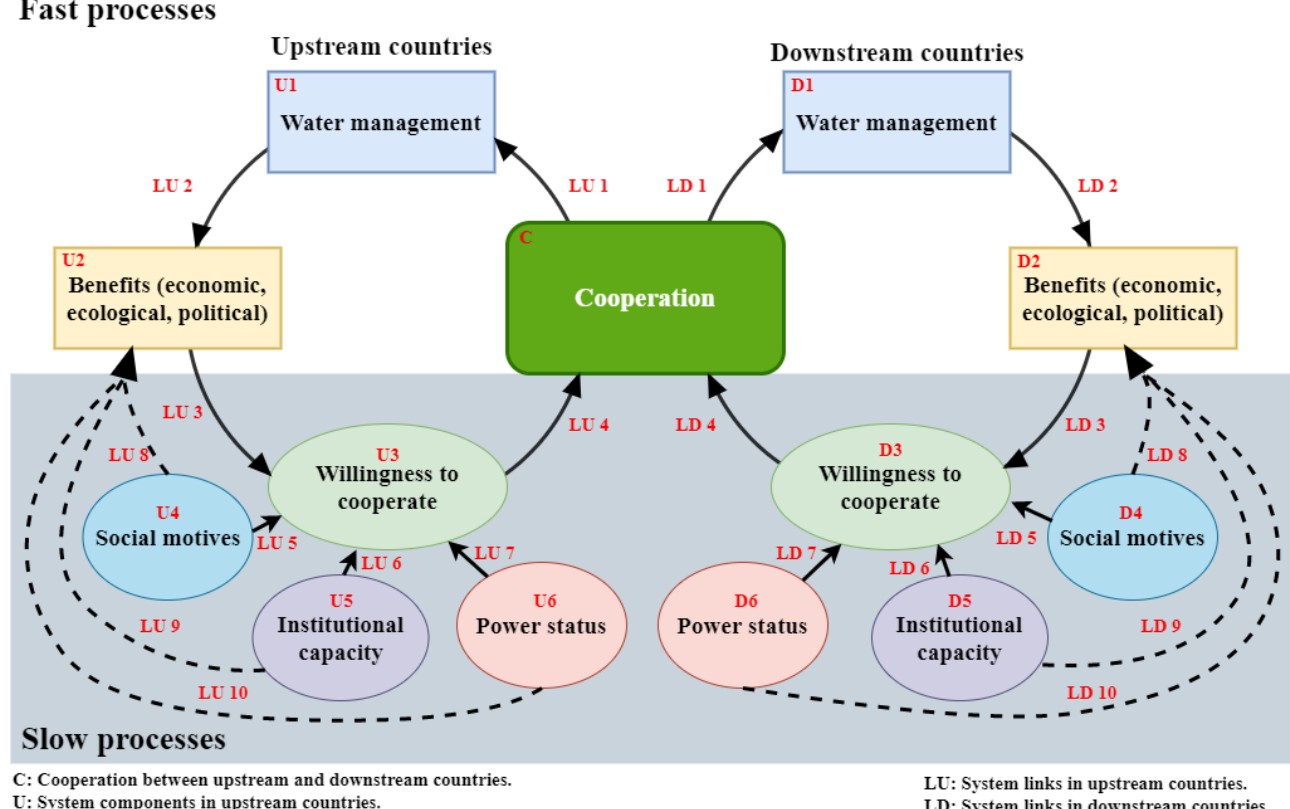

**Figure 1. A social-hydrological framework for understanding cooperation and conflict in transboundary rivers.**

Willingness to cooperate is a slow process influenced by both slow and fast processes. In the fast processes, it is directly influenced by the benefits one country will potentially receive, including short-term and direct economic benefits, long-term ecological benefits, and indirect political benefits that reflect the reality of water management in transboundary benefits. These benefits will be achieved through change in water management, e.g., changing the dam storage and then streamflow. In the slow processes, the willingness to cooperate is influenced by social motives, power status, and institutional capacity. When

receiving benefits from the river, social motives determine how one country perceive their benefits, i.e., the weighting they exert on different kinds of benefits (economic, ecological, political). Riparian countries that are pro-self in nature are more likely to put heavier weighting on individual economic benefits, while pro-social countries would value more on long-term collective ecological benefits. Geographical location and economic/political power also impact the extent to which riparian countries are willing to cooperate, i.e., countries with countervailing economic and political power are far more likely to reach

cooperation; the geographical locations (the spatial dependent level), can also have a direct influence on the willingness to cooperate. Institutional capacity, a path-dependent social variable, indicates the adaptive capacity that can manage and maintain the cooperation status. The strength of this variable indicates the extent to which riparian countries follow their agreed



provisions or cooperation. Furthermore, feedbacks between the change in social motives, power status and institutional capacity and change in comprehensive benefits, which are functions of change in hydrology, are recognised in this framework.

With these feedbacks the unintended and undesired outcomes as emergent phenomena can be observed and explained. It should be noted that changes in cooperation and willingness to cooperate occur in both domestic and international contexts. These contextual factors include climate change, natural and human disasters, population growth, urbanisation, change in sovereignty, diplomacy and national security, and change in national boundary.

This framework, by bringing the slow and hidden societal processes into existing hydrological process-based model on

transboundary water management, changes cooperation as a categorical variable (0,1) to a continuous process variable (0-1). This change enables observations of the change of cooperation status and societal processes underlying it, and development of formal models to simulate feedbacks between change in societal processes and change in hydrology through the benefits functions. Thus, this socio-hydrological framework can explain the unintended and undesired outcomes and contributes to understanding of the mechanism that drives cooperation between riparian countries. To make this framework concept to act as

a 'middle ground' between the meta-level concepts and a specific model driven by a context, the section below will provide a general set of variables and relationships between them from which analysts can choose a subset and further specify according to a specific problem and the system being investigated.

### 3.2 Framework specification

To implement the framework introduced above, variables reflecting each system component should firstly be defined and

measured in a quantitative way. We list the definitions and measures of these variables to our best knowledge (Table 1).

**Table 1. The definition and measure of the variables in framework concept.**

| System component | Variables and definition | Measure |
|---|---|---|
| Water management | Water supply (dam storage) and water management: dam operation (water release). <br> Water demands. | Directly obtained from hydrological gauge stations or simulation. <br> Water demand varies from sector to sector. |
| Benefits | Economic benefits include hydropower, flood control, irrigation, fishing, and others. <br> Ecological benefits include those at catchment, in stream and floodplains. <br> International political benefit is a comprehensive reputation of a country in the world. | These benefits are functions of their water demands. <br> They are derived based on their respective disciplines (neoclassical economics, ecology and international politics). |





| Cooperation | Change in existing water sharing agreement or treaty among riparian countries, a status variable. | A Boolean variable (0 or 1). |
|---|---|---|
| Willingness to cooperate | A latent process variable reflecting the dynamic process of cooperation. | A continuous variable between 0 and 1. It is a function of benefits, social motives and power status and institutional capacity. When Willingness to cooperate reaches 1, Cooperation is 1, otherwise it is 0. |
| Social motives | Value reflection of different countries on cooperation. It influences the weightings among different benefits and reflects the different conditions for cooperation. There are different types of motives for cooperation. | Measured as an index of 0-1. It is developed based on cognitive psychology and cultural sociology. |
| Power status | A variable expresses social-economic ranking of a country in the world and the geographical location of this country in a transboundary river. | A comprehensive index. While many datasets reflecting global social-economic development index and power are available, selection of these datasets should be based on international politics. |
| Institutional capacity | A path-dependent variable, reflecting the adaptive capacity to absorb systems changes. | Various indicator sets have been developed in literature to reflect the differences of institutional capacity, but selection of these datasets should be based on institutional economics. |

Obviously, to observe and measure the characteristic variables of the societal system is a big challenge. This includes quantifying the international political benefits, social motives, power status, and institutional capacity. Social motive (value) is the primary driver of change in water sharing among riparian countries, but there is no recognised method to represent and quantify it as discussed in the previous section. In the existing socio-hydrological models, it remains ad hoc and only anonymous societal variables or representative indicators are used due to the absence of long-term observations of human behaviour (Di Baldassarre et al., 2019). The availability of 'big data' has provided an unprecedented opportunity to analyse and model the complex structures and dynamics in the societal systems (Bhattacharya & Kaski, 2019). However, the context-site specific and unstructured data and "thick descriptive" approach in the social sciences have to be generalized, structured and quantified before they can be integrated into hydrological models (Wei et al., 2018; Newig & Rose, 2020; Olsen, 2004). We suggest neither replacing nor abandoning critical interpretive and more holistic qualitative research but integrating the more "explanatory" ones into the hydrological model. This will be achieved by transforming narratives into quantitative explanatory data through a content coding scheme, which is rooted in a context-mechanism-outcome configurations and allows for triangulation by multiple data sources (Pawson & Tilley, 1997). We, with this approach, have tracked the evolution of societal value on water with media data for different research contexts (Wei et al., 2017; Xiong et al., 2016), Wu et al., 2018; Wei et al., 2021; Geneva, 2021).





Functions between societal variables and hydrological variables and between societal variables then need to be established, in particular those functions to reflect the feedbacks is an even bigger challenge. For examples, how power status, institutional capacity, and social motives individually or collectively influence the willingness to cooperate? And how do they individually

or collectively give feedback to change in hydrological condition (water management) through change in weighting of different benefits? Although there are rich qualitative and descriptive understandings of the interactions among these societal variables and their links with biophysical changes in social sciences, it remains a challenge to formalize these relationships in a systemic, consistent, and generalizable way. We suggest that the empirical relations in case studies based on the types of change trend of these variables should be firstly developed (Sterman, 2001; Pentland, 2015). With enough understandings from the inductive

perspective, some more theoretical formulations can be established.

Following that, these societal variables need to be calibrated with the societal data. It is recognised as a weakness in existing social-hydrological models that the societal components (e.g., represented by environmental awareness or community sensitivity) were not directly calibrated with social data (Di Baldassarre et al., 2019). There are many existing societal data available for model calibration, including global databases and indicator-based assessment on conflict and cooperation

discussed in the previous section, also those datasets reflecting global social-economic development index, power, and reputation (Treverton & Jones, 2015). We see that to calibrate the conflict and cooperation in the transboundary rivers provide an opportunity to improve the development of socio-hydrological models in general. Finally, model uncertainty should be noted as the transboundary river is a complex adaptive system which is characterized by non-linearity, heterogeneity, multiple equilibrium states and cross-scale dynamics. We may not be able to make predictions of cooperation in the traditional sense

and the conventional sensitivity analysis may not perfectly fit for this kind of social-hydrological model. Rather, projections on possible future trends may be useful to inform future transboundary river management (Srinivasan et al., 2017).

Several points are given here for use of this framework:. 1) this general set of variables help prevent analysts from missing key variables which could lead to insufficient representation of reality in hydro-economic models. 2) analysts can customise a subset from this general set of variables and relationships between them based on the issues at hand. 3) This set of variables

and relationships between them do not always change in different evolutional phases of conflict and cooperation dynamics in a transboundary river (i.e. some links in Figure 1 do not always change with time).

## 4 Case studies: Application of the framework to specific transboundary rivers

We use the Columbia River, the Mekong River, and the Nile River, three well-known transboundary rivers, as case studies to demonstrate the applicability of this proposed framework (Figure 2). The evolutionary dynamics of conflict and cooperation

in these transboundary rivers will be described according to their development stages, and the framework variables that are of significant consideration in each case will be discussed (Table 2). The formalized modelling based on this social-hydrological framework on these three case rivers are reported in this issue as separate research papers.



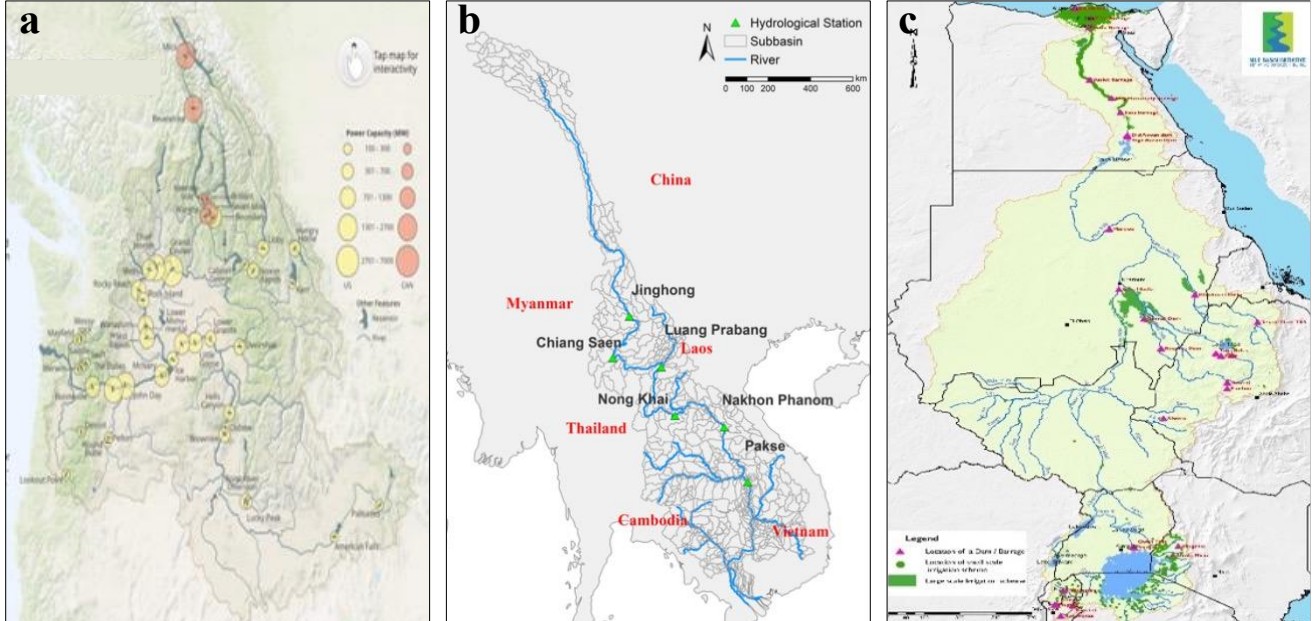

**Figure 2. Case examples: (a) Columbia River (USACE, 2020), (b) Lancang-Mekong River (Lu et al., 2020), (c) Nile River (NBI, 2014).**

## 4.1 The Columbia River

The Columbia River starts in British Columbia and has a basin that extends 670,807 km². The basin covers seven U.S. states (Washington, Oregon, Idaho, Wyoming, Montana, Nevada, and Utah) and drains to the Pacific Ocean via Oregon. Only 15% of the river's length flows through Canada, but the Canadian portion accounts for 38% of the average annual flow. The river has multiple domains of use: hydropower, fishing, irrigation, recreation, navigation, and ecosystem uses. Millions of people in the Pacific Northwest rely on these services. The river has high volume and large seasonal variability of flow. Downstream areas face significant flood risks because of strong seasonality of flow and spring snowmelt peaks. The evolution of conflict and cooperation in the Columbia River can be divided into three stages.

At Stage I (~early 1960s), development increased along the river in Washington and Oregon. Strong seasonality of flow and spring snowmelt peaks posed significant threats and caused damages. In 1948, flooding driven by snowmelt and heavy rainfall breached the levee and destroyed Vanport, Oregon's second largest city, as well as Trail, B.C. It caused dozens of deaths and extensive property damage in both the U.S. and Canada (corresponding to U2 and LU3, D2 and LD3 in Figure 1). These floods were the impetus for the U.S. to seek cooperation with Canada (U3 and LU4, D3 and LD4). The U.S. found it difficult to capture enough water to control flood levels within its portion of the river. At the same time, more than 90 percent of the potential damages in the basin are in the downstream portion of the river.



At Stage II (early 1960s ~ early 1990s), joint studies began after the 1948 flooding to explore possible storage sites in Canada and analyse the benefits of sharing river between the countries (U2 and LU3, D2 and LD3). It was concluded that cooperation benefits are more advantageous to both sides compared to options available through individual operation (U3 and LU4, D3 and LD4). Following negotiations, the Columbia River Treaty was completed in 1964 to manage the river for the joint benefit

of both countries, focusing on flood control and hydropower (C). Under this agreement, the U.S. paid Canada $64.4 million to rent 8.45 million acre feet of storage space in Canada (LU2 and LD2). These funds were used to build and operate three large storage dams (Keenleyside, Mica, and Duncan) on the Canadian side and the Libby Dam on the U.S. side. Canadian dams must be operated to lower reservoir levels and provide storage space during spring and summer seasons to capture water upstream to prevent flooding (U1 and D1). In addition, the U.S. pays Canada 50% of the projected U.S. power benefit generated

by Canadian storage, also known as the "Canadian Entitlement", for the expected avoidance of flood damages through 2024 (LU2 and LD2). In exchange, the controlled release of these dams provided an opportunity for more efficient hydropower production in the downstream because of more predictable and flexible flows (U1 and D1). The cooperation through the Treaty has been used as a pinnacle for international cooperation on non-navigational water uses.

At Stage III (early 1990s ~ present), changing socio-environmental conditions have altered the context of the 1964 treaty (LU3

and LD3). Urban development, such as the City of Portland, along the downstream portion of the river that has increased the value at risk. Also, tribal groups and First Nations whose existence depend on the river have suffered loss of fish (salmons and steelhead) from dam construction. They requested their sovereignty right (cultural and natural resources) to be respected (U4 and LU5, D4 and LD5). Thirteen species of anadromous salmon, steelhead, and sturgeon are listed under the Endangered Species Act (ESA). By the 1990s, salmon and steelhead populations reached alarmingly low levels, prompting aggressive

action at the Federal level to impose stronger regulations on dam operators to adjust their operating strategies to support the recovery of fish (U5 and LU10, D5 and LD10). The primary operational change is that hydropower operators must augment seasonal river flows and increase spill at dams to assist downstream migration of juvenile fish, decrease water temperature, and increase flow velocity (U1 and D1). Spills occur when hydropower operators divert some portion of the river flow, particularly in spring and summer, away from the hydropower turbines, which allows for fish to pass the dam without risking

injury. However, hydropower producers experience financial losses because these spills utilize water that could otherwise be used to produce hydropower (LU2 and LD2). At the same time, the U.S. continues to pay the same Canadian Entitlement agreed upon in the Treaty, which has created the perception of decreased hydropower benefit on the U.S. side (U2 and D2). The U.S. entity calculated that the value of Canadian storage and downstream power value should be around $26 million USD in electricity (about 1/10th of the estimated worth of the Canadian Entitlement) because it does not consider fishery needs,

agriculture, non-Treaty dams, and annual variability in precipitation (LU3 and U3). Canada, on the other hand, argues that the value provided by Canadian storage is much higher than the current Entitlement (e.g., additional benefits of navigation, recreation, irrigation, and fisheries), and that additional costs should be borne by the U.S (LD3 and D3). These different arguments from Canada and the U.S. will be base for renegotiations on cooperation beyond 2024 (LU4 and LD4).



### 4.2 The Mekong River

The Mekong River Basin (known as the Lancang-Mekong River Basin in China) spans 795,000 km$^2$ across six countries (China, Myanmar, Thailand, Laos, Vietnam, and Cambodia) in South-East Asia with over 60 million populations. It is one of the largest and longest transboundary rivers and also one of the most productive inland fisheries in the world (MRC, 2018; Yorth, 2014). About 85% of the basin's population lives in rural areas, whose livelihoods and food are highly dependent on the river system (FAO, 2011). Conflict and cooperation in the Mekong Basin mainly evolved around constructions of large

dams and water distribution (De Stefano et al., 2017; Wei et al., 2021), which demonstrated five stages from 1999 to 2018 (Lu et al., 2021 and Wei et al., 2021).

    Stage I (1999 ~ 2003) was characterised by limited conflict in the basin due to absence of dam construction (Yorth, 2014). "Agreement on the cooperation for the Sustainable Development of Mekong River Basin" was signed by all members in the Mekong River Commission (Hirsch and Cheong, 1996) (C). Riparian countries shared the economic benefits from the Mekong

River, in particular agricultural and fishery development provided high economic returns to the downstream countries (Lu et al., 2021) (LD2). Stage II (2004 ~ 2005) was characterised by unexpected hydrological changes due to the severe droughts. The changes in the hydrological systems of all riparian countries were beyond the agreement in Stage I, which led to increased conflict among riparian countries as the economic benefits from agriculture and fishery reduced significantly for downstream countries (LU3 and LD3). Cooperative demand peaked for both upstream and downstream countries in 2005 (Wei et al., 2021).

At Stage III (2006 ~ 2009), China agreed to provide hydrological information of the Mekong River to improve understanding of changes in the upstream hydrological systems (Yorth, 2014) (C). The volume of cargo trade from China to downstream also increased to provide additional economic benefits to downstream countries (LU2 and LD2).

    Stage IV (2010 ~ 2016) was featured by rapid construction of dams, leading to changes in the hydrological and ecological systems. Upstream countries (i.e. China and Laos) had strong interests in hydropower development to increase their domestic

economic benefits (U4 and LU8). China started to construct the Xiaowan dam in 2010 and the Nuozhadu dam in 2012. The downstream hydrological changes resulted from these upstream dam constructions included increase in dry season runoff and reduction of runoff peak in the flood season (Hoanh et al., 2010). Vietnam censured China for increasing salinization and degradation of the downstream ecological system (Youth et al., 2014). Severe droughts in 2015 and 2016 further reduced the economic benefits from fishery and agriculture for the downstream countries. The losses of fishery benefit were about USD

162 million in 2015 (D4 and LD8). This aggravated concerns and criticisms of downstream countries against upstream countries (D2 and LD3). During Stage V (2017 ~ present), the impacts of ecological degradations from last stage were recognised by all riparian countries and most countries' cooperative willingness increased (Wei et al., 2021) (LU4 and LD4). China regarded the geopolitical values and diplomatic relations as an important international political benefit (Urban et al., 2018) in addition to economic benefits (U6 and LU10), therefore more willing to cooperate with other riparian countries (Lu

et al., 2021) (LU7). Major hydropower projects had been completed and several treaties and plans were signed towards cooperation (Wei et al., 2021) (C).





### 4.3 The Nile River

The Nile River with an estimated length of over 6800 km is one of the longest and greatest rivers in the world. It covers about 10.3% of the African continent and has a total population of about 250 million people. The river is shared by 11 countries: the
stakes and interests of Egypt, Sudan and Ethiopia are classified as very high and those of Uganda, Tanzania, Kenya, Burundi and Rwanda, Eritrea, South Sudan and the Democratic Republic of Congo as low. The conflict and cooperation dynamics in the Nile River management demonstrated four stages.

At Stage I (1956 ~ 1989), Egypt and Sudan reached bilateral agreement in 1959 to divide the Nile water between the two countries with hydraulic infrastructure in place (refer to Agreement between the Republic of the Sudan and the United Arab
Republic for the full utilization of the Nile waters) (C). The exclusive rights to utilize the Nile waters enabled huge economic benefits and bonus of hydropower for Egypt (Allan, 1999) (LD2 and D2), which largely impacted other countries' socio-economic developments due to their limited access and rights to use the water (Kameri-Mbote 2007) (LU 2 and U2). In addition, in 1973 and 1984-1985, major droughts stroke Ethiopia killing millions of people, which raised Ethiopia's awareness of its needs to develop the Nile waters (Gebrehiwot et al, 2011) (U4 and LU5). In stage II (1989 ~ 1998), Ethiopia started to ask for
transboundary cooperation and sharing the waters of the Nile (U3). Negotiation and lobbying were common but up until the end of 1990s (LU4 and LD4), the willingness to cooperate remained elusive (U3 and D3). This was because Egypt remained the most powerful riparian country capable of influencing the hydro-political interactions across the basin (D6 and LD7), while other countries exhibited weak capacity to change their status due to their limited capacity to exert power at both regional and international levels (Cascão, 2009; Cascão & Nicol, 2016) (U6 and LU7).

At Stage III (1999 ~ 2010), new cooperation process initiated, which unfolded into two parallel tracks (C). The technical track, Nile Basin Initiative (NBI), started as a temporary initiative to manage transboundary issues; and the policy track to drive negotiation toward Cooperative Framework Agreement (CFA) (Cascão & Nicol, 2016). The riparian countries established new cooperative norms through joint activities under a Shared Vision Program (SVP) and two Subsidiary Action Programmes (SAPs), one for the Eastern Nile (ENSAP) and one for the Nile Equatorial Lakes respectively (NELSAP). ENSAP and
NELSAP, through multiple projects promoted the joint identification and planning of hydraulic projects that would bring tangible benefits to these countries (Cascão & Nicol, 2016) (LU2 and LD2). Joint Multipurpose Project (JMP), started in 2005, reached the stalemate in 2009, while the upstream countries decided to sign the CFA in 2010. External financial support for the JMP decreased and Ethiopia realized that the direct economic benefits it gained from the projects were limited, regardless the growing economic needs between 2000 and 2010 in Ethiopia (U4 and LU5). At the same time the Arab Spring started in
Egypt and signalled the decline of its political stability (which causes foreign investments in Egypt to further decline to zero) (D6 and LD7). As a result of both indirect and unintended consequences, the multilateral cooperation failed (failure of current C). At Stage IV (2011 ~ present), Ethiopia stated its intention to construct the Grand Ethiopia Renaissance Dam (GERD) (U6 and LU7). Sudan also recognized the benefits of the GERD and necessity of expanding irrigation due to the 2008 food crisis (D4 and LD5), making it more willing to cooperation for joint water management in the Nile (U3 and D3). Sudan has now



shifted from siding with Egypt to being more open to cooperation with Ethiopia (LU4 and LD4). Agreement has been made for Sudan to buy electricity from Ethiopia once the dam is finished and to potentially gain water for irrigation as well (C).

Based on the evolutionary dynamics of conflict and cooperation in these transboundary rivers described above, the variables important to each case and their status are summarized in Table 2. It can be seen that the Columbia River provides a successful case so far for transboundary water management although there emerge changes in benefit distributions between the riparian

countries that require further negotiations for cooperation. Sharing the same societal values, respecting and appreciating each country's power and rights, and strong institutional capacities (both physical and legal/political) are preconditions for success. The Mekong River provides a complex case for conflict and cooperation among six countries with their respective benefits, diverse cultural and international political backgrounds. This case demonstrates that inclusion of economic, ecological, international political benefits is crucial to understand conflict and cooperation dynamics, while at the same time recognizing

the different institutional capacities in different countries. The Nile River provides a hydrological and economic case on conflict and cooperation among riparian countries. The unstable institutional capacities and strong power differences were the root cause for the strong conflict and weak cooperation in the region. Quantifying the power status and their relationships with change in hydrology are crucial in this case.

**Table 2. Site-specific characteristics of the three case rivers in application of the framework.**

| System component | Site-specific characteristics | | |
| --- | --- | --- | --- |
| | **The Columbia river** | **The Mekong river** | **The Nile river** |
| Water management | Water management: dam operation. | Development of dam storage and water management (dam operation). | Development of dam storage. |
| Benefits | Economic benefits: hydropower, flood control. Ecological benefits: protection of salmon. | Economic benefits: hydropower, flood control, irrigation, fishing. Ecological benefits: preventing from downstream salinization. International political benefit. | Economic benefits: irrigation and hydropower. |
| Cooperation | Existence of Treaty, due to renew in 2024. | No formal treaty or agreement existing for all riparians with only regional agreement and basin – wide cooperation initiative. | Existence of formal bilateral agreement, few basin-wide cooperative norms, which have all stopped functioning. |





| Willingness to cooperation | Higher end of the range of [0, 1]. | Fluctuating between the range of [0, 1]. | Lower end of the range of [0,1]. |
|---|---|---|---|
| Social motives | Homogeneous with minor difference. | Highly different due to different cultural background. | Homogeneous with little difference. |
| Power status | Almost equivalent. | Upstream countries with stronger comprehensive power. | Downstream countries with stronger comprehensive power. |
| Institutional capacity | Very high in both hard and soft institution in both countries. | Moderate – level. | Very weak in all riparian countries. |


## 5 Conclusion

The social-hydrological framework proposed in this paper brings the slow and hidden societal processes into integrated hydrological and economic models and establishes the feedbacks between changes in societal variables and hydrological cycles via benefit functions. It contributes to revealing the mechanism that drive conflict and cooperation and the development of

socio-hydrology by improving representation and measurement of societal variables. Furthermore, this meta-theoretical framework can act as a platform integrating the advances in understanding of conflict and cooperation from multiple disciplines including hydrology, ecology, economics, sociology, and political sciences.

As demonstrated in the application of this framework in the Nile, Mekong, and Columbia rivers in this paper and other formalized modelling papers in this special issue, this framework will provide a common language and consistent way for

comparative analysis of conflict and cooperation dynamics in the over 300 transboundary rivers. This analysis will assist explanation of why a socio-hydrologic phenomenon occurs in one case but not in others, identification of different modes of cooperation in transboundary rivers thus building a global database of management strategies for more economically, ecologically and politically sustainable transboundary rivers.

## Acknowledgement

The presented work was developed within the framework of the Panta Rhei Research Initiative of the International Association of Hydrological Sciences (IAHS). We acknowledge the support from the 2019 Summer Institute on Socio-hydrology and Transboundary Rivers held in Yunnan University, China. In particular, we would like to give our special thanks to Professor Amin Elshorbagy from University of Saskatchewan, Professor Giuliano Di Baldassarre from Uppsala University, Professor Günter Blöschl from Vienna University of Technology, Professor Marco Borga, University of Padova, Assistant Professor



Margaret Garcia from Arizona State University and Associate Professor Megan Konar from University of Illinois at Urbana-Champaign.

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
