# Peer review of "A socio-hydrological framework for understanding conflict and cooperation in transboundary rivers"

_Hydrology and Earth System Sciences, 2021_

## Referee Comment (RC2)

Review of "A socio-hydrologic framework for understanding conflict and cooperation in transboundary rivers"
Yongping Wei et al, 2021 (HESSD)

This paper describes a socio-hydrological framework for understanding cooperation in transboundary basins. The authors first introduce the importance of transboundary basins along with the benefits of a "meta-theoretical" framework to understand transboundary cooperation. The authors identify disciplines that study transboundary basins but are limited in understanding or modeling the causes of transboundary cooperation. The authors use this gap in understanding to motivate and then describe the framework, after which the authors use the framework to understand transboundary dynamics in three major transboundary basins: the Columbia, Lancang-Mekong, and Nile river basins. The framework itself emphasizes the hidden variable "willingness to cooperate" as a central component of transboundary cooperation, and describes how other socio-economic components of the transboundary system influence this variable.

This type of framework can be valuable for understanding conflict and cooperation transboundary basins, both in terms of conceptualizing cooperation within individual case studies as well as comparing and contrasting across transboundary basins. Given the importance of transboundary basins to water supplies around the world, such research is highly important. With that said, revisions are needed to clarify the contribution of this paper and provide an evidence basis with which to evaluate the framework. My concerns include that (a) the review of existing literature on conflict and cooperation is overly general, (b) I don't see where the authors articulate their approach to developing the framework, and (c) the case studies (and possibly the framework) are built upon other manuscripts that have also been sent to the same special issue in HESS. It is therefore important that the authors more clearly delineate the objective and contribution of this manuscript (including differentiating it from similar works recently sent to HESS).

I elaborate on these concerns below.

1. The abstract states: "This article aims to review the existing knowledge on conflict and cooperation in transboundary rivers from a multidisciplinary perspective…"
   a. However, the literature review of other disciplines (Section 2.2) is cursory, and more specifics would provide a clearer picture of how the framework relates to other fields. Ideally the literature review would not only motivate the framework but also provide a theoretical foundation for the framework. It seems to me that the literature review focuses on *quantitative models* of conflict and cooperation in transboundary basins, which is quite different from a *general* literature review on conflict and cooperation in general. The issue with this is that the framework presented seems to be quasi-quantitative – it is packaged in such a way that it could be formulated into a quantitative model, but seeks to find a middle ground whereby simplifications needed for simulation. To given an example, the paragraph beginning on line 104 reads:

      "Neoclassical economics has dominated the simulation and explanation of human cooperation behaviour. It explains cooperation in riparian countries from a purely economic perspective, focusing on the tangible outcomes received by these countries, assuming them as rational actors with perfect information about all potential choices

and their consequences (Schill et al., 2019). Hydrological models have been integrated with neoclassical economic models to simulate cooperation in transboundary rivers by optimizing the incremental economic benefits under a set of specific societal constraints. Thus, the influences from the social dimension are only considered as residuals from explanations of rational economic behaviour. These models have been criticized for being overly simplistic, and unable to capture the diversity of human behaviour (Schlüter et al., 2017), and thus fail to reflect the reality of conflict and cooperation in transboundary rivers (Wei et al., 2021)."

This paragraph would fit within a general discussion of sociohydrology, but I'm concerned it oversimplifies the contributions/relationship between neoclassical economics and transboundary studies. Further, it does not articulate how neoclassical economics informs the development of the transboundary framework. I find most of Section 2.2 to be similarly general and lacking details that would be expected from a general literature review of conflict and cooperation in transboundary watersheds. I would encourage the authors to more clearly describe which authors/manuscripts used which models (based theory from neoclassical economics), rather than use neoclassical economics as the subject (as in the first two sentences of the paragraph), which is confusing to me because the theory is *applied by* researchers to explain cooperation.

2.  The epistemological basis of the framework is not clear. Even though it is presented as a "proposed" framework, describing the origins of the framework (e.g., the methods / theoretical foundation) is critical because this will shape how the framework should be interpreted and applied. Line 190 states: "We … thus develop this framework by following the system theory in particular the complex system theory. The development of this framework are also built on the recent advances on understanding the coupled human relationships from sociale-nvironment ecological system (Folke et al, 2005), the Coupled Human and Nature Systems (CHANS) (Liu et al, 2007) and the general socialhydrological framework (Elshafei et al, 2014)." But these citations are particularly general and more evidence should be provided to support the framework. Related to this concern are the following points:
    a.  Is the framework meant to be general? What does it capture and what does it miss?
    b.  The notion that "Social motives," "Institutional capacity," and "Power status" affect international cooperation is uncontroversial and well established. These concepts are broadly defined, and therefore the relationship with "Willingness to cooperate" is likely context specific – with that said, Table 1 indicates that each of these can be computed via index. The rationale behind this choice should be more clearly explained along with a description of the relationship between these variables and "willingness to cooperation". This rationale would also make it easier to evaluate the structure of the framework.
    c.  Watershed management is motivated strongly by interests within countries, but there is no arrow from "benefits" to "water management" in Fig 1.
    d.  The only interaction between countries is through the binary variable "Cooperation", which is itself influenced by the "willingness to cooperate" of each individual country. Cooperation between countries is typically not binary — it can be continuous and it can be multi-dimensional, including many areas of cooperation beyond water or transboundary resources — the choice for a single binary variable is therefore confusing to me.

e. Additionally, what about the relational aspects that influence willingness to cooperation vis-a-vis specific countries. For instance, Sudan appears to have maintained a high willingness to cooperate, but this fact conceals an underlying shift in preference to cooperate from Egypt towards Ethiopia. Part of this shift was driven by changing power differentials across the three countries and the relational aspect of this differential must be considered, but does not appear to be reflected in the framework. Additionally, how do bilateral relations factor into the framework, and is this exogenous or endogenous?

3. Some aspects of the framework are unclear. For instance:
   a. Why are some variables slow or fast? Willingness to cooperate is marked as a slow variable but it could change rapidly with, e.g., a newly elected political leader.
   b. Willingness to cooperate is driven by "Social motives," "Institutional capacity," and "Power status". These variables can be represented by indices, but it's unclear how these indices could be related to changes in willingness to cooperate. For instance, the social motives variable is represented by an index in the range 0-1. But how does this index relate to cooperation, and why?

4. This paper presents three cases that build upon other manuscripts in the same special issue of HESS (p 281). These papers should all be cited on L281 and the authors should be clearer (up front, ie the abstract/introduction) about the relationship between this manuscript and the other case studies, including how this paper builds on those studies (e.g., was the framework developed based on those studies?) and what specifically this paper introduces that is a new contribution to the literature. I appreciate the value of using the framework to compare across case studies in Table 2. With that said, the case studies were described in such a way to fit within the framework, but it's unclear what value the framework added to understanding the individual case studies.

---

## Author Comment (AC1)

We greatly appreciate the positive comments from the Referee #1. Here we address those queries and concerns which are very constructive and highly valuable.

*I believe the paper would benefit by bringing it all together a bit more. The synthesis table and how the case studies are similar or different in terms of various elements of the framework such as social motives, power, institutional capacity etc is useful but how such system components talk to each other is not clear from the narratives of the case studies.*

**Agreed.**

We will revise the purpose of Section 4 as follows:

"We use the Columbia River, the Mekong River, and the Nile River, three well-known transboundary rivers, as case studies to demonstrate the applicability of this proposed framework (Figure 2). This framework adds values to the case studies by identifying the key variables and key links between variables that are crucial to understand the evolutionary dynamics of conflict and cooperation in these transboundary rivers, and influence stage transitions in these rivers. It will provide basis for developing formalized socio-hydrological models (Table 2)."

We will revise the section paragraph by paragraph to reflect similar or different in terms of various elements of the framework among case studies. We will add a summary paragraph to summarize these differences among the three case studies as follows:

"It is seen that the Columbia River provides a successful case so far for cooperation in transboundary rivers although there emerge changes in benefit distributions between the riparian countries that require further negotiations for cooperation. Sharing the same societal values, appreciating each country's power and rights, and strong institutional capacities (both hard and soft) are major drivers for success. The Mekong River provides a complex case for conflict and cooperation among six countries with their respective benefits, and diverse cultural and international political backgrounds. This case demonstrates that inclusion of economic, ecological, international political benefits is crucial to understand conflict and cooperation dynamics while recognizing the different institutional capacities in different countries. The Nile River provides an unsuccessful case of which unstable institutional capacities and unfavourable asymmetric power distributions were the root cause for strong conflict and weak cooperation. Therefore, the framework can identify key variables and links that explains conflict and cooperation in transboundary rivers."

*Description of slow and fast dynamics is not so clear in the case studies. While the authors argue that hydro-economic treatment of transboundary river sociohydrology has gaps, they do not convincingly demonstrate that these gaps are filled by bringing in additional components through the case studies. I also think that the authors are unclear about how to 'quantify' various variables and concepts corresponding to these system components (also the sets of these variables and components appear to be 'open' sets) - they do allude to somethings in the paper (Table 1) but it is not clear to me how it is educating the slow and/or fast dynamics. Perhaps a more tangible effort to quantify some slow/fast dynamic equations (such equations can be conceptual in nature) will help. Also, some more tangible evidence of how some of its corresponding variables can be observed/measured, e.g. through behavioural experiments or surveys in ennvironmental psychology will help. Finally, what is it that neo-classical economics cannot explain that the proposed system components help explain in the narratives of the case studies? Almost all of the case studies can be explained by dynamic non-coperative game theory under uncertainty (evolving benefits, power, institutions, capacity and their feedbacks under exogenous shocks). So, what exactly the framework is accomplishing remains unclear and should be clearly brought forward. What is endogenous, what is exogenous to the system, how behavioral experiments/ environmental psychology data collection and analysis methods are being deployed, is it very slowly evolving culture/institutions and its effect on norms, perception of risk and capacity (given the time horizon of the case studies discussed) etc that are not covered by the current hydroeconomic models and needed to fully make sense of the presented narratives of the three basins?*

**Agreed.**

As the reviewer's comments are high level and quite comprehensive, we will rewrite Section 3.1 - 3.2 as follows:

[revised manuscript text omitted]

*I also think the figures and language at places can be improved.*

**Agreed.**

Our apologies for the grammar errors and improper use of language. We will carefully revise the whole manuscript.

---

## Author Comment (AC2)

We greatly appreciate the positive comments from the Referee #2. Here we address those queries and concerns which are very constructive and highly valuable.

With that said, revisions are needed to clarify the contribution of this paper and provide an evidence basis with which to evaluate the framework. My concerns include that (a) the review of existing literature on conflict and cooperation is overly general, (b) I don't see where the authors articulate their approach to developing the framework, and (c) the case studies (and possibly the framework) are built upon other manuscripts that have also been sent to the same special issue in HESS. It is therefore important that the authors more clearly delineate the objective and contribution of this manuscript (including differentiating it from similar works recently sent to HESS). I elaborate on these concerns below.

1. The abstract states: "This article aims to review the existing knowledge on conflict and cooperation in transboundary rivers from a multidisciplinary perspective..."

a. However, the literature review of other disciplines (Section 2.2) is cursory, and more specifics would provide a clearer picture of how the framework relates to other fields. Ideally the literature review would not only motivate the framework but also provide a theoretical foundation for the framework. It seems to me that the literature review focuses on quantitative models of conflict and cooperation in transboundary basins, which is quite different from a general literature review on conflict and cooperation in general. The issue with this is that the framework presented seems to be quasi-quantitative – it is packaged in such a way that it could be formulated into a quantitative model, but seeks to find a middle ground whereby simplifications needed for simulation. To give an example, the paragraph beginning on line 104 reads:

"Neoclassical economics has dominated the simulation and explanation of human cooperation behaviour. It explains cooperation in riparian countries from a purely economic perspective, focusing on the tangible outcomes received by these countries, assuming them as rational actors with perfect information about all potential choices and their consequences (Schill et al., 2019). Hydrological models have been integrated with neoclassical economic models to simulate cooperation in transboundary rivers by optimizing the incremental economic benefits under a set of specific societal constraints. Thus, the influences from the social dimension are only considered as residuals from explanations of rational economic behaviour. These models have been criticized for being overly simplistic, and unable to capture the diversity of human behaviour (Schlüter et al., 2017), and thus fail to reflect the reality of conflict and cooperation in transboundary rivers (Wei et al., 2021)."

This paragraph would fit within a general discussion of sociohydrology, but I'm concerned it oversimplifies the contributions/relationship between neoclassical economics and transboundary studies. Further, it does not articulate how neoclassical economics informs the development of the transboundary framework. I find most of Section 2.2 to be similarly general and lacking details that would be expected from a general literature review of conflict and cooperation in transboundary watersheds. I would encourage the authors to more clearly describe which authors/manuscripts used which models (based theory from neoclassical economics), rather than use neoclassical economics as the subject (as in the first two sentences of the paragraph), which is confusing to me because the theory is applied by researchers to explain cooperation.

**Agreed.**

We will delete the sentence "to review the existing knowledge on conflict and cooperation in transboundary rivers from a multidisciplinary perspective" in the abstract.

We will revise the aim of this paper in the final paragraph of Section Introduction as "Contributing to the filling of knowledge gaps between multidisciplinary (in particular social sciences) linkages with hydrology is the objective of this study. This will be done through three steps. First, an overview of the existing literature on conflict and cooperation in transboundary rivers from multiple disciplines and the integration of social sciences with hydrology will be provided. This will provide an understanding of the preliminary concepts in a wide range of disciplines on cooperation and conflict and identify the gaps of their linkage with hydrology."

We will revise Section 2.2 with more detailed description on the specific models, for example, the hydrology-economic model.

2. The epistemological basis of the framework is not clear. Even though it is presented as a "proposed" framework, describing the origins of the framework (e.g., the methods / theoretical foundation) is critical because this will shape how the framework should be interpreted and applied. Line 190 states: "We ... thus develop this framework by following the system theory in particular the complex system theory. The development of this framework are also built on the recent advances on understanding the coupled human relationships from social-environment ecological system (Folke et al, 2005), the Coupled Human and Nature Systems (CHANS) (Liu et al, 2007) and the general socialhydrological framework (Elshafei et al, 2014)." But these citations are particularly general and more evidence should be provided to support the framework. Related to this concern are the following points:

a. Is the framework meant to be general? What does it capture and what does it miss?

b. The notion that "Social motives," "Institutional capacity," and "Power status" affect international cooperation is uncontroversial and well established. These concepts are broadly defined, and therefore the relationship with "Willingness to cooperate" is likely context specific – with that said, Table 1 indicates that each of these can be computed via index. The rationale behind this choice should be more clearly explained along with a description of the relationship between these variables and "willingness to cooperation". This rationale would also make it easier to evaluate the structure of the framework.

c. Watershed management is motivated strongly by interests within countries, but there is no arrow from "benefits" to "water management" in Fig 1.

d. The only interaction between countries is through the binary variable "Cooperation", which is itself influenced by the "willingness to cooperate" of each individual country. Cooperation between countries is typically not binary — it can be continuous and it can be multidimensional, including many areas of cooperation beyond water or transboundary resources — the choice for a single binary variable is therefore confusing to me.

e. Additionally, what about the relational aspects that influence willingness to cooperation visa-vis specific countries. For instance, Sudan appears to have maintained a high willingness to cooperate, but this fact conceals an underlying shift in preference to cooperate from Egypt towards Ethiopia. Part of this shift was driven by changing power differentials across the three countries and the relational aspect of this differential must be considered, but does not appear to be reflected in the framework. Additionally, how do bilateral relations factor into the framework, and is this exogenous or endogenous?

**Agreed.**

As the reviewer's comments are very comprehensive, we will rewrite Section 3.1 - 3.2 as follows:

3.1 The framework concept

We develop a meta-theoretical framework to address the knowledge gaps in understanding conflict and cooperation in transboundary rivers which are identified in the section above. This framework will act as a 'middle ground' between the meta-level concepts and theories from related disciplines as introduced above and specific models driven by a particular context/a specific problem for building an interdisciplinary bridge to study the mechanism that drives conflict and cooperation in transboundary rivers.

We develop this framework based on the complex adaptive system theory and recent advances on understanding the coupled human-environment relationships from social-ecological systems (Folke, 2006), the Coupled Human and Nature Systems (CHANS) (Liu et al, 2007) and the social-hydrological framework (Elshafei et al, 2014). A complex adaptive system is of non-linearity, heterogeneity, multiple equilibrium states and cross-scale dynamics to present emergent behaviours. Specifically, we consider transboundary rivers as complex adaptive systems comprising water management (hydrological), ecological, economic, cultural, institutional, and political subsystems in each riparian country (Figure 1, demonstrating a case involving two riparian countries). These subsystems co-evolve, each affecting the others in each riparian country in a long timeframe. It is widely recognised in the co-evolutionary processes, hydrological and economic variables are of "fast" characteristics which work at the scale of seconds to years, and ecological and societal variables are relatively "slow" which often work at the scale of decades to centuries. Those slow variables (subsystems) often show a pattern of "punctuated equilibrium" (Eldredge & Gould, 1972) characterized by a long period of stasis being punctuated by a more rapid change that disrupts the equilibrium. For example, the 'cultural (societal value) lag' is well noted in the literature (Rosenschöld et al., 2014). Power status sometimes could not change for decades, even several thousands of years in ancient periods, but it could change suddenly through an elected political leader in modern times. It is the interaction of 'fast' processes and 'slow' processes that determine the system thresholds which, if crossed, cause the system to move into a new state (Sivapalan et al., 2012).

In this framework, cooperation (whether to cooperate or not) occurs as the emergent behaviour between subsystems among riparian countries, which is a result of non-linear responses and multiple feedbacks between these subsystems (Figure 1). In conventional hydrology-economic models, whether to cooperate or not is defined as a binary variable (0, 1) to examine the evolutionary dynamics of cooperation. It only involves the fast processes indicated in upper part of Figure 1. As the cooperation continues, the value of cooperation will always be 1. It only involves the fast processes: water management conditions, the resultant benefits, and their direct feedbacks as indicated in the upper part of Figure 1. The slow processes that influence the cooperation decision in each riparian country's system are largely neglected. This framework extends the existing understanding of cooperation from integrated hydrology-economic models to include the willingness to cooperate, a hidden variable representing the slow societal processes (as the processes in lower part of Figure 1).

C: Cooperation between upstream and downstream countries

U: System components in upstream countries.

D: System components in downstream countries

LU: System links in upstream countries. LD: System links in downstream countries. Dash lines indicate feedback links.

[revised manuscript text omitted]
 Mekong river during 1991-2018 by using cumputer-based sentiment mining in the newpapers collected in the Factiva, which is published in the same issue (Wei et al., 2020).

Functions between societal variables and hydrological variables and between societal variables then need to be developed. It is obvious that the stronger the social motives for cooperation, the higher the willingness to cooperate. The stronger the institutional capacity, the higher the willingness to cooperate. However, the power status may behave differently. Stronger power status can have positive or negative influences on the willingness to cooperate, depending on the direction of social motives. For example, China, which is located upstream of the Mekong River (geographical strength) and has stronger economic/political power than other riparian countries, but it does not always positively support cooperation. The functions between these variables are often expressed in a logit form (Hofbauer and Sigmund, 2003). However, we suggest that the relations between these variables and existing qualitative and descriptive understandings of the interactions among these variables in social sciences (Pentland, 2015; Sterman, 2001). With enough understandings from the inductive perspective, some more theoretical formulations can be established.

Following that, these societal variables need to be calibrated with the societal data. It is recognised as a weakness in existing social-hydrological models that the societal components (e.g., represented by environmental awareness or community sensitivity) were not directly calibrated with societal data (Di Baldassarre et al., 2019). There are many existing societal data available for model calibration, including global databases and indicator-based assessment on conflict and cooperation discussed in the previous section, also those datasets reflecting global social-economic development index, power, and reputation (Treverton & Jones, 2015). We see that to calibrate the conflict and cooperation in the transboundary rivers provide an opportunity to improve the development of socio-hydrological models in general. Finally, model uncertainty should be noted as the transboundary river is a complex adaptive system which is characterized by non-linearity, heterogeneity, multiple equilibrium states and cross-scale dynamics. We may not be able to make predictions of cooperation in the traditional sense and the conventional sensitivity analysis may not perfectly fit for this kind of social-hydrological model. Rather, projections on possible future trends may be useful to inform future transboundary river management (Srinivasan et al., 2017).

In a word, this framework, by bringing the slow and hidden societal processes into existing hydrologyeconomic models on transboundary rivers, understand the cooperation from a binary variable (0, 1) underlying the fast processes to a continuous process between (0-1) with combination of cooperation and willingness to cooperate underlying the interaction between fast processes and slow processes. It enables observations of the change of cooperation status and societal processes underlying it for development of formal models to simulate feedbacks between change in social processes and change in hydrology through the benefit functions. Thus, this socio-hydrological framework can explain the unintended and undesired outcomes and contributes to understanding of the mechanism that drives cooperation between riparian countries. Compared to the existing hydrology-economic models with the game theory, it mechanistically and quantitatively explains residuals from explanations of rational economic behaviour (uncertainly), thus provide more precise and comprehensive knowledge on conflict and cooperation management in transboundary rivers.

**3. Some aspects of the framework are unclear. For instance:**

a. WhFiny are some variables slow or fast? Willingness to cooperate is marked as a slow variable but it could change rapidly with, e.g., a newly elected political leader.

b. Willingness to cooperate is driven by "Social motives," "Institutional capacity," and "Power status". These variables can be represented by indices, but it's unclear how these indices could be related to changes in willingness to cooperate. For instance, the social motives variable is represented by an index in the range 0-1. But how does this index relate to cooperation, and why?

**Agreed.**

Please see the rewritten Sections 3.1 - 3.2 provided in Question 2.

4. This paper presents three cases that build upon other manuscripts in the same special issue of HESS (p 281). These papers should all be cited on L281 and the authors should be clearer (up front, ie the abstract/introduction) about the relationship between this manuscript and the other case studies, including how this paper builds on those studies (e.g., was the framework developed based on those studies?) and what specifically this paper introduces that is a new contribution to the literature.

**Agreed.**

We will make clear the relationship between this manuscript and the other case studies. As a matter of fact, among all published papers in the issue, only the paper entitled *"Socio-hydrologic modelling of the dynamics of cooperation in the transboundary Lancang-Mekong River*, written by You Lu, Fuqiang Tian, Liying Guo, Iolanda Borzi, Rupesh Patil, Jing Wei, Dengfeng Liu, Yongping Wei, David Yu, and Murugesu Sivapalan (Hydrol. Earth Syst. Sci., 25, 1883–1903, 2021. doi: https://doi.org/10.5194/hess-25-1883-2021) was written based on the framework proposed in this manuscript, although it was published a little bit earlier. It should be noted that most of authors of this manuscript were the authors of that published paper.

I appreciate the value of using the framework to compare across case studies in Table 2. With that said, the case studies were described in such a way to fit within the framework, but it's unclear what value the framework added to understanding the individual case studies.

The primary purpose of applying the framework in the three case studies in this manuscript is to demonstrate the applicability of the proposed framework. This framework adds values to the individual case studies by identifying the key variables and key links between variables that are crucial to understand the evolutionary dynamics of conflict and cooperation in these transboundary rivers, and influence stage transitions in these rivers. It will provide basis for developing a formalized socio-hydrological model. The paper *"Socio-hydrologic modelling of the dynamics of cooperation in the transboundary Lancang-Mekong River"* (Hydrol. Earth Syst. Sci., 25, 1883–1903, 2021. doi: <a href="https://doi.org/10.5194/hess-25-1883-2021">https://doi.org/10.5194/hess-25-1883-2021</a>) is a good example that the authors identified the key variables and developed functions based on the descriptive and qualitative analysis above. We will make these explanations clear in the revised manuscript.

---

## Author Response (AR1)

**Response to Reviewers**

Dear Editors and Reviewers,

Thank you very much for your time and effort for reviewing the article titled "**A socio-hydrological framework for understanding conflict and cooperation in transboundary rivers**".

We really appreciate all of your insightful comments. To address them, please see a revised version of the manuscript (changes marked in red font). We have also provided point-by-point responses to each of your comments in blue font below:

**Referee #1:**

*I believe the paper would benefit by bringing it all together a bit more. The synthesis table and how the case studies are similar or different in terms of various elements of the framework such as social motives, power, institutional capacity etc is useful but how such system components talk to each other is not clear from the narratives of the case studies.*

**Agreed.** We have revised the purpose of Section 4 in **Line 290-295**, and the whole section paragraph by paragraph to reflect similarities or differences in terms of various elements of the framework among case studies **(Line 300-415)**. A summary paragraph has been added in **Line 415-425** to summarize the implications of the proposed framework to the three case studies**.**

*Description of slow and fast dynamics is not so clear in the case studies. While the authors argue that hydro-economic treatment of transboundary river sociohydrology has gaps, they do not convincingly demonstrate that these gaps are filled by bringing in additional components through the case studies. I also think that the authors are unclear about how to 'quantify' various variables and concepts corresponding to these system components (also the sets of these variables and components appear to be 'open' sets) - they do allude to somethings in the paper (Table 1) but it is not clear to me how it is educating the slow and/or fast dynamics. Perhaps a more tangible effort to quantify some slow/fast dynamic equations (such equations can be conceptual in nature) will help. Also, some more tangible evidence of how some of its corresponding variables can be observed/measured, e.g. through behavioural experiments or surveys in ennvironmental psychology will help. Finally, what is it that neo-classical economics cannot explain that the proposed system components help explain in the narratives of the case studies? Almost all of the case studies can be explained by dynamic non-coperative game theory under uncertainty (evolving benefits, power, institutions, capacity and their feedbacks under exogenous shocks). So, what exactly the framework is accomplishing remains unclear and should be clearly brought forward. What is endogenous, what is exogenous to the system, how behavioral experiments/ environmental psychology data collection and analysis methods are being deployed, is it very slowly evolving culture/institutions and its effect on norms, perception of risk and capacity (given the time horizon of the case studies discussed) etc that are not covered by the current hydroeconomic models and needed to fully make sense of the presented narratives of the three basins?*

**Agreed.** As the reviewer's comments are high level and quite comprehensive, we have fully rewritten Section 3.1 - 3.2 **(Line 185-290)**.

*I also think the figures and language at places can be improved.*

Our apologies for the grammatical errors and improper use of language. We have carefully revised the whole manuscript and updated both Figure 1 and 2.

**Referee #2:**

*With that said, revisions are needed to clarify the contribution of this paper and provide an evidence basis with which to evaluate the framework. My concerns include that (a) the review of existing literature on conflict and cooperation is overly general, (b) I don't see where the authors articulate their approach to developing the framework, and (c) the case studies (and possibly the framework) are built upon other manuscripts that have also been sent to the same special issue in HESS. It is therefore important that the authors more clearly delineate the objective and contribution of this manuscript (including differentiating it from similar works recently sent to HESS). I elaborate on these concerns below.*

*1. The abstract states: "This article aims to review the existing knowledge on conflict and cooperation in transboundary rivers from a multidisciplinary perspective…"*

> *a. However, the literature review of other disciplines (Section 2.2) is cursory, and more specifics would provide a clearer picture of how the framework relates to other fields. Ideally the literature review would not only motivate the framework but also provide a theoretical foundation for the framework. It seems to me that the literature review focuses on quantitative models of conflict and cooperation in transboundary basins, which is quite different from a general literature review on conflict and cooperation in general. The issue with this is that the framework presented seems to be quasi-quantitative – it is packaged in such a way that it could be formulated into a quantitative model, but seeks to find a middle ground whereby simplifications needed for simulation. To give an example, the paragraph beginning on line 104 reads:*

> *"Neoclassical economics has dominated the simulation and explanation of human cooperation behaviour. It explains cooperation in riparian countries from a purely economic perspective, focusing on the tangible outcomes received by these countries, assuming them as rational actors with perfect information about all potential choices and their consequences (Schill et al., 2019). Hydrological models have been integrated with neoclassical economic models to simulate cooperation in transboundary rivers by optimizing the incremental economic benefits under a set of specific societal constraints. Thus, the influences from the social dimension are only considered as residuals from explanations of rational economic behaviour. These models have been criticized for being overly simplistic, and unable to capture the diversity of human behaviour (Schlüter et al., 2017), and thus fail to reflect the reality of conflict and cooperation in transboundary rivers (Wei et al., 2021)."*

> *This paragraph would fit within a general discussion of sociohydrology, but I'm concerned it oversimplifies the contributions/relationship between neoclassical economics and transboundary studies. Further, it does not articulate how neoclassical economics informs the development of the transboundary framework. I find most of Section 2.2 to be similarly general and lacking details that would be expected from a general literature review of conflict and cooperation in transboundary watersheds. I would encourage the authors to more clearly describe which authors/manuscripts used which models (based theory from neoclassical economics), rather than use neoclassical economics as the subject (as in the first two sentences of the paragraph), which is confusing to me because the theory is applied by researchers to explain cooperation.*

**Agreed.** We have revised the abstract **(Line 15 – 20)** and the final paragraph of Section Introduction **(Line 60 – 70)** to clarify the aim of this paper.

We have also revised Section 2.2 with more detailed description on the specific models, for example, the hydrological-economic model **(Line 105 – 115)**.

*2. The epistemological basis of the framework is not clear. Even though it is presented as a "proposed" framework, describing the origins of the framework (e.g., the methods / theoretical foundation) is critical because this will shape how the framework should be interpreted and applied. Line 190 states:*

*"We ... thus develop this framework by following the system theory in particular the complex system theory. The development of this framework are also built on the recent advances on understanding the coupled human relationships from social-environment ecological system (Folke et al, 2005), the Coupled Human and Nature Systems (CHANS) (Liu et al, 2007) and the general socialhydrological framework (Elshafei et al, 2014)." But these citations are particularly general and more evidence should be provided to support the framework. Related to this concern are the following points:*

> *a. Is the framework meant to be general? What does it capture and what does it miss?*

> *b. The notion that "Social motives," "Institutional capacity," and "Power status" affect international cooperation is uncontroversial and well established. These concepts are broadly defined, and therefore the relationship with "Willingness to cooperate" is likely context specific – with that said, Table 1 indicates that each of these can be computed via index. The rationale behind this choice should be more clearly explained along with a description of the relationship between these variables and "willingness to cooperation". This rationale would also make it easier to evaluate the structure of the framework.*

> *c. Watershed management is motivated strongly by interests within countries, but there is no arrow from "benefits" to "water management" in Fig 1.*

> *d. The only interaction between countries is through the binary variable "Cooperation", which is itself influenced by the "willingness to cooperate" of each individual country. Cooperation between countries is typically not binary — it can be continuous and it can be multi-dimensional, including many areas of cooperation beyond water or transboundary resources — the choice for a single binary variable is therefore confusing to me.*

> *e. Additionally, what about the relational aspects that influence willingness to cooperation vis-a-vis specific countries. For instance, Sudan appears to have maintained a high willingness to cooperate, but this fact conceals an underlying shift in preference to cooperate from Egypt towards Ethiopia. Part of this shift was driven by changing power differentials across the three countries and the relational aspect of this differential must be considered, but does not appear to be reflected in the framework. Additionally, how do bilateral relations factor into the framework, and is this exogenous or endogenous?*

**Agreed.** As the reviewer's comments are very comprehensive, we have fully rewritten Section 3.1 - 3.2 **(Line 185-290)**.

*3. Some aspects of the framework are unclear. For instance:*

> *a. WhFiny are some variables slow or fast? Willingness to cooperate is marked as a slow variable but it could change rapidly with, e.g., a newly elected political leader.*

> *b. Willingness to cooperate is driven by "Social motives," "Institutional capacity," and "Power status". These variables can be represented by indices, but it's unclear how these indices could be related to changes in willingness to cooperate. For instance, the social motives variable is represented by an index in the range 0-1. But how does this index relate to cooperation, and why?*

**Agreed.** Please see the rewritten Section 3.1 - 3.2 **(Line 185-290)**.

*4. This paper presents three cases that build upon other manuscripts in the same special issue of HESS (p 281). These papers should all be cited on L281 and the authors should be clearer (up front, ie the abstract/introduction) about the relationship between this manuscript and the other case studies, including how this paper builds on those studies (e.g., was the framework developed based on those studies?) and what specifically this paper introduces that is a new contribution to the literature.*

**Agreed**. We have clarified in **Line 420 – 425** the relationship between this manuscript and the case studies.

*I appreciate the value of using the framework to compare across case studies in Table 2. With that said, the case studies were described in such a way to fit within the framework, but it's unclear what value the framework added to understanding the individual case studies.*

We have clarified the purpose of applying the framework and how it can add values to the three case studies in **Line 290 – 295**, and **Line 410 – 425**.

---

## Author Response (AR2)

**Report #1:**

Thank you for your response. I still find table 1 and the text abstract.

Thank you. To improve the quality of this manuscript,

1) We have revised Section 1 to make clear the knowledge gap of this manuscript and the angle and purpose of this manuscript;

2) We have revised Section 2 guided by the more clearly stated purpose in Section 1. Particularly, to improve the links between this section on literature and next section on development of a socio-hydrological framework, we have added a new Table (Table 1) to summarize the current understandings on conflict and cooperation in transboundary rivers and their contribution to developing the conceptual framework. In addition, we have corrected those imprecise statements and added new literature.

3) The revised Section 1 and Section 2 as mentioned above make Table 2 and the text in Section 3 easier to understand; and

4) We have revised the old table 1 (Table 2 in the new manuscript) and the text of whole Section 3 to make them clearer and serve to the revised purpose in Section 1.

In addition,

5) We have restructured Section 4 to make the purpose of this section clearer; and

6) We have heavily rewritten Section Conclusion and Section Abstract to more precisely reflect the key findings of this manuscript.

To make it more tangible for the audience of HESS, as also commented in the previous round, perhaps the authors may want to present some conceptual equations underlying the framework. This would help hydrologists to associate with important measurable variables. The paper https://agupubs.onlinelibrary.wiley.com/doi/full/10.1002/2015WR017896 does a really good job in doing something similar and may help the authors in understand what I mean.

Thanks. We have substantially improved the sub-section on the multiple relationships between the sub-systems described in the old table 1 (new Table 2). Specifically,

1) We have stated that those well-developed models can provide the basis for developing the formal model based on the proposed framework (**Line 245-250**):

"As descried in Section 2, there are well developed integrated hydrology-ecology-geomorphology models and hydrology-economics models. The general guidelines for developing the social-hydrological models and mathematically specifying those fast and slow processes have been well developed in the literature (e.g. Elshafei et al, 2014 and 2015; Sivapalan and Bloeschl, 2015)".

2) We have discussed the function form and possible temporal stages of the three societal variables: social motive (value), institutional capacity and power status which are the core part of this framework according to the relevant theory (**Line 250-265**):

"It is widely recognised that many societal changes are gradual processes in time following a sigmoid function (S-shaped curve) (e.g. Choi et al., 2015; Ghanbarnejad et al., 2014). We adopted the transition theory on societal evolution by Rotmans et al. (2001) and Rotmans (2005) (Figure 2), which identified a predevelopment phase when the current status quo remains for the system, a take-off phase when the process of change becomes visible as the state of the system begins to shift, an acceleration phase when visible structural changes occurs relatively rapidly, and a stabilization phase when the societal system change stabilizes. Societal transitions can fail in any of these phases, indicated by a backlash or a lock-in situation, and the whole system may even collapse when uncertainties and risks of chaos are too high. Thus for each of social motive, institutional capacity, and power status, we can consider their temporal developments in the form of a sigmoidal function (Hofbauer and Sigmund, 2003) (Eq.1):

$$S_i(t) = a + \frac{k}{1+e^{-t}} \tag{1}$$

where $S_i(t)$ is the societal dynamics in time t, with i representing social motive, institutional capacity, and power status, a and k are the constant values representing the scale and rates of development in time, and e is the Euler's number."

[Figure]

**Figure 2. Stages and possible pathways of development of societal system (adopted from Rotmans et al., 2001; Rotmans, 2005).**

3) We conceptually formulated the most important relationship in the framework, i.e. the relationship between the willing to cooperate and three societal variables (social motive (values), institutional capacity and power status), meanwhile, we recognize that the functional relations should be investigated in different case studies based on the types of dynamics of these variables and existing qualitative and descriptive understandings of the interactions among these variables in social sciences **(Line 265-280):**

"It is obvious that the stronger the social motive and institutional capacity for cooperation, the higher the willingness to cooperate. However, stronger power status can have positive or negative influences on the willingness to cooperate, depending on the directions of social motive. For example, China, which is located upstream of the Lancang-Mekong River (geographical strength) and has stronger economic/political power than other riparian countries, but it does not always positively support cooperation. The conceptual function between the willingness to cooperate and the three societal variables can be written as (Eq.2):

$$Willingness\ to\ cooperate(t) = f\{S_{social\ motives}(t)^{g[S_{institutional\ capacity}(t),S_{power\ status}(t)]}\} \qquad (2)$$

where f is a power function chosen to consider social movie as the primary driver (i.e., base of the power function) for cooperation in comparation to institutional capacity and power status; g is the index function reflecting the parallel importance of institutional capacity and power status to willingness to cooperate. However, we suggest that the relations between these variables in different case studies should be investigated based on the types of dynamics of these variables and existing qualitative and descriptive understandings of the interactions among these variables in social sciences as described in Section 2 (Sterman, 2001; Pentland, 2015). With enough understandings from the inductive perspective, more theoretical formulations can be established."

Also in this context , a clear response to a comment from the previous version would be very much appreciated "how behavioral experiments/ environmental psychology data collection and analysis methods are being deployed, is it very slowly evolving culture/institutions and its effect on norms, perception of risk and capacity (given the time horizon of the case studies discussed) etc that are not covered by the current hydroeconomic models and needed to fully make sense of the presented narratives of the three basins."

Thanks. A review of how behavioural experiments/environmental psychology data collection and analysis methods are being deployed has been added **(Line 220-235):**

"It can be seen from Table 2 that the measurement of social motives (values) is a big challenge in the framework, which is also a common challenge for developing socio-hydrological models (Di Baldassarre et al., 2019). The commonly adopted methods for value measure are surveys, experiments, and in-depth interviews and participant observation. Surveys, which contain survey items on value that participants are asked to rate along a 9-point (or less) scale, is an important part of the methodological repertoire for values research. However, it may be subject to measurement error due to the discrepancy between how people respond to surveys and how they actually behave

(Schwartz,1992). The experimental approach such as cooperation in games is powerful as it measures actual behaviours, but it has less external validity and generalizability (how well the results generalize to situations outside the experiment and how well the subjects in the experiment represent the general population) (McClintock, 1978). In-depth interviews and participant observation has the advantage of uncovering how people are articulating their values rather than asking them to react to survey items, but this approach is labour intensive and also difficult to generalize across studies (Diez et al, 2015). In addition, all these methods are often cross-sectional in time or only reflect the value change in a short timeframe, thus cannot meet the longitudinal (decades or longer) requirement for simulating complex adaptive systems. Recently, the importance of discourse in changing values have been emphasized as communication with other individuals shapes and reshapes the emphasis we place on values (Habermas, 1991)."

A clearer statement of our method has been made **(Line 235-245):**

"The availability of 'big data' (e.g. media) has provided an unprecedented opportunity to analyse and model the complex structures and dynamics in the societal systems (Bhattacharya & Kaski, 2019). We have developed an approach to integrate "thick descriptive" societal data into hydrological models by transforming narratives into quantitative data through a content coding scheme which is rooted in a context-mechanism-outcome configurations and allows for triangulation by multiple data sources (Pawson & Tilley, 1997; Wei et al., 2018; Newig & Rose, 2020; Olsen, 2004). With this approach, we have tracked the evolution of societal value on water with media data under different research contexts (Wei et al., 2017; Xiong et al., 2016; Wu et al., 2018). In transboundary rivers, we quantitively tracked the societal values on conflict and cooperation of the riparian countries in the Lancing-Mekong River during 1991-2018 which is published in the same issue (Wei et al., 2021)."

The application of our method in the Lancang-Mekong River which is published has been mentioned for readers' further interest.

**Report #2:**

The authors have re-written large portions of the manuscript. Overall I find the motivation and description of the framework considerably improved. The framework should provide needed structure to sociohydrological discussions of transboundary basins, in particular social aspects of transboundary cooperation. This is an important advancement.

One possible concern with the paper is that a number of statements lack precision and could be confusing. I have highlighted a couple below, but otherwise would leave this to the authors' discretion.

Thanks. We have carefully gone through the whole manuscript and corrected those statement lack of precision or being confusing.

I still also have the following minor comments: I don't see improvement in Section 2.2 and find my previous concerns unaddressed. In the context of the other improvements, this can be considered a minor issue. With that in mind, however, I read this section as a literature review that is not directly linked to the message of the paper and with a number of statements that lack precision or are misleading. For instance, one sentence that should be improved is (line 106): "Neoclassical economics has dominated the simulation and explanation of human cooperation behavior." This isn't true, for instance, Elinor Ostrom's work has been used to understand cooperative behavior for decades. Another sentence in the paragraph about behavioral economics (line 118): "In transboundary rivers, whether people choose to cooperate or not relies on one country's expectations on absolute economic benefits, their benefits in previous periods as a reference level, relative gains compared to other countries, and intangible benefits such as ecological, social, political, or diplomatic benefits." Are the authors arguing that these are the drivers of transboundary cooperation, or stating how behavioral economics might view transboundary cooperation?

Thanks. We have heavily rewritten Section 2.2. Specifically:

1) We have more clearly stated the purpose of this overview in last paragraph of Introduction, which is "the existing literature on conflict and cooperation in transboundary rivers is overviewed, which provides the constituent disciplinary and empirical basis for developing such a conceptual framework" **(line 55-60)**;

2) Guided by the more clearly stated purpose, we have revised the whole **Section 2.2**;

3) To improve the links between this section on literature and next section on development of a socio-hydrological framework, we have added a new Table (**Table 1**) to summarize the current understandings on conflict and cooperation in transboundary rivers and their contribution to developing the conceptual framework; and

4) We have corrected those imprecise statements and add new literature including those mentioned by the reviewer (particularly **Line 100-115**).

The relationship between the framework and the Columbia River is implicit in Section 4.1. The relationship between the key hidden variables and cooperation in the three stages should be explicitly described (or hypothesized) within the section, not just summarized in Table 2. As written, the case study reads as a summary and timeline of events, rather than an application of the framework. The other case studies should also include more explicit references to the key hidden variables from the framework.

Thanks. We have restructured this section and revised the corresponding paragraph to improve its logic. Please see the whole **Section 4.2**. The revised logic of this section is stated as: " We use the Columbia River, the Lancang-Mekong River, and the Nile River, three well-known transboundary rivers, as case studies to demonstrate the applicability of this proposed framework (Figure 3). We will firstly narrate the evolutionary dynamics of conflict and cooperation in these transboundary rivers according to their development stages, then use Figure 2 and Table 2 to identify the key sub-systems from the narratives of each case river to see if the framework can grasp the core dynamics of conflict and cooperation in these transboundary rivers." **(Line 290-300)**

Line 225: "Social motives are the primary driver of willingness to cooperate". Suggest changing to "a primary driver"? Otherwise please better defend this statement.

Thanks. It has been changed. Please see **Line 270-275**.

---

## Author Response (AR3)

Dear Prof. Di Baldassarre,

Thank you very much for your comment. We have updated the statement as: **"There are 286 rivers around the world that cross the boundaries of two or more countries (TWAP, 2022)."**

Supporting reference has been added:

[Dataset] TWAP (2022). Transboundary Waters Assessment Programme - River Basins Component. Available: http://twap-rivers.org/.

Best regards,

Yonping Wei

On behalf of the author group.